# In-Context Generalization to New Tasks From Unlabeled Observation Data

## Abstract

Large pretrained models in natural language processing and computer vision have achieved impressive capabilities by training on vast internet-scale corpora. However, for sequential decision-making agents, such as robots and other autonomous systems, it is difficult and expensive to collect large amounts of expert demonstrations hindering their ability to learn new tasks efficiently. Leveraging unannotated internet videos as a resource, we propose an approach to train a generalist agent capable of few-shot adaptation to new tasks without fine-tuning. Our method, Prompt-DTLA, learns a *latent* action model to annotate video sequences with latent actions that enables training an in-context causal transformer policy on these annotated trajectories. At inference, the agent can generalize to new, unseen tasks using few-shot *in-context* demonstrations without additional fine-tuning. Prompt-DTLA offers a potential solution for scaling robot learning with free, internet-scale data rather than expensive human demonstrations, enabling generalist agents to learn new tasks from unlabeled data sources.

## 1. Introduction

Large pretrained foundations models in natural language processing (Brown et al., 2020; Touvron et al., 2023) and vision (Dosovitskiy et al., 2021; Minderer et al., 2022) have demonstrated impressive capabilities in a variety of challenging downstream tasks. The success of such models can be attributed to the scalability of the Transformer architecture (Vaswani et al., 2017) and the abundance of diverse internet-scale pretraining text and image corpora, which enable these models to capture rich, context-aware representations.

In robotics, there have been efforts to collect large amounts of data to train generalist agents (Padalkar et al., 2023). However, this process is usually very labor intensive and may risk damaging expensive hardware. Can sequential decision-making agents, such as robots, leverage the wealth of readily available internet videos? Videos often provide visual demonstrations of how to perform various tasks and could be used to facilitate learning from freely available internet data in a similar way to natural language processing and computer vision.

Traditional algorithms to solve decision-making problems, such as reinforcement learning, rely on low-level action labels in order to train agents. In this work, we propose an approach for leveraging unannotated video data to train a generalist agent capable of few-shot adaptation to a new task without any model updates or fine-tuning. To overcome the lack of explicit action supervision, we learn an inverse dynamics model (IDM) for latent actions in an unsupervised manner (Bruce et al., 2024) using the video dataset. We use this IDM to annotate sequences of consecutive observations with latent action information. We then train an agent capable of in-context learning on trajectories containing latent actions and observations. At inference time, our in-context agent receives few-shot expert demonstrations as conditioning. Without extra fine-tuning, we produce a policy capable of performing the desired task. We present our framework, **Prompt**-**D**ecision **T**ransformers with **L**atent **A**ctions (Prompt-DTLA) for learning a generalist agent from purely video data.

## 2. Preliminaries

**Offline Meta-RL.** In standard RL, the goal is to learn an optimal policy for a single task. This is inefficient as each new task requires training a completely new policy. The goal of meta-RL (Finn et al., 2017) is to design a learning agent that can adapt to new, unseen tasks with few expert demonstrations. A meta-RL agent learns over a distribution $p(\mathcal{T})$ of possible environments or *tasks* during meta-training. Each task $\mathcal{T}_i \sim p(\mathcal{T})$ is described by an MDP $\mathcal{M}_i = \langle \mathcal{S}, \mathcal{A}, \mathcal{R}_i, \mathcal{P}_i, \mu_i, \gamma \rangle$, where $\mathcal{S}$ and $\mathcal{A}$ are the state and action spaces, $\mathcal{R}$ is a reward function, $\mathcal{P}$ is the transition function, $\mu$ is the initial state distribution, and $\gamma$ is a discount factor. Typically, $\mathcal{S}$, $\mathcal{A}$, and $\gamma$ are shared across tasks, while $\mathcal{R}_i$ and $\mathcal{P}_i$ are task-specific. More recently, meta-RL has been extended to the offline setting (Mitchell et al., 2021;

---

[1]Anonymous Institution, Anonymous City, Anonymous Region, Anonymous Country. Correspondence to: Anonymous Author <anon.email@domain.com>.

Preliminary work. Under review by the 1st In-context Learning Workshop at the International Conference on Machine Learning (ICML). Do not distribute.

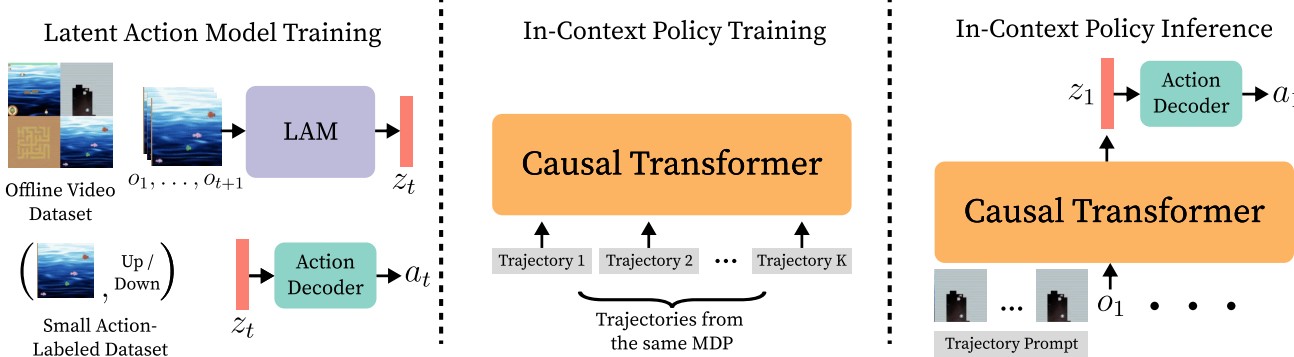

Figure 1: **(Left) Latent Action Model Training**. We assume access to a large offline corpus of observation-only trajectories across a diverse suite of tasks. We learn a latent action model which, given the observation history and the next observation, $(o_1, \ldots, o_t, o_{t+1})$, predicts the latent action at timestep $t$. Using a few action-labeled trajectories, we train a decoder that maps latent actions to environment actions. **(Middle) DTLA Training.** We train an autoregressive causal transformer with multiple trajectory sequences from the same MDP concatenated together, where the latent actions between consecutive observations are labeled using the pre-trained LAM. **(Right) DTLA Inference.** We evaluate our in-context policy using few-shot expert trajectories as prompts. Our ICL policy outputs latent actions, which are decoded into environment actions using the learned decoder.

Dorfman et al., 2021; Xu et al., 2022), reducing the need for expensive online data collection in meta-training and adaptation by using available offline datasets. The objective of offline meta-RL is to learn a policy on the training tasks, $\mathcal{T}_{train}$, that efficiently maximizes performance on a new task sampled from the task distribution, $\mathcal{T}_{test}$, typically after some additional task specific fine-tuning. Prompt-DTLA tackles the problem setting of offline meta-RL.

**In-context Learning for Meta-RL.** Large language models in NLP have been shown to have *in-context learning* (Brown et al., 2020) capabilities: They can perform a new task simply by conditioning on a few training examples in their context. The in-context learning paradigm is attractive because the model can learn a new task without any additional fine-tuning or parameter updates. Prompt-DT (Xu et al., 2022) translates in-context learning to the RL domain. The authors extend the decision transformer (Chen et al., 2021) architecture, which frames decision-making as a sequence modeling problem of next token prediction, to take few-shot *trajectory prompts*. By doing offline training on a distribution of tasks, Prompt-DT learns to generalize to new test tasks using only few-shot data. However, Prompt-DT assumes access to action and reward labeled offline datasets which are often expensive and difficult to collect. Prompt-DTLA can learn from observation-only data while leveraging the in-context ability of transformers to generalize to unseen test tasks.

**Latent Action Models.** Most RL algorithms rely on training data that include action labels. Recent works, LAPO (Schmidt & Jiang, 2023) and Genie (Bruce et al., 2024), however, have shown that it is possible to learn *latent actions*

in a completely unsupervised way from purely observational data. The main difference between these two works is that LAPO considers only a small window of previous observations while Genie attend to the full trajectory context using a Transformer. Additionally in LAPO, the number of discrete codes $K \gg |A|$ while in Genie, $K = |A|$, where $|A|$ is the number of ground truth actions. Both of these prior works only consider learning policies for single MDPs and do not focus on the challenge of generalization to unseen tasks.

## 3. Prompt Decision Transformers with Latent Actions

We introduce Prompt-Decision Transformers with Latent Actions (Prompt-DTLA), an approach for training generalist agents from purely observation data. The key idea of Prompt-DTLA is to learn a latent action model (LAM) from a large, unlabelled offline video dataset for annotating our data with latent actions and then train a Transformer agent capable of in-context generalization using few-shot trajectory prompts. An overview of the different components of Prompt-DTLA is shown in Figure 1.

**Latent Action Model.** We train a latent action model (LAM) following prior work (Schmidt & Jiang, 2023; Bruce et al., 2024). The input to the LAM is a sequence of $t + 1$ observations, $o_{1:t+1} = (o_1, \ldots, o_t, o_{t+1}) \in \mathbb{R}^{T \times H \times W \times C}$. The LAM consists of an *inverse* dynamics model (IDM), $f_\phi(z_t \mid o_{1:t+1})$, which outputs the latent action at time $t$ conditioned on all previous observations and the next observation, and a *forward* dynamics model (FDM), $g_\psi(o_{t+1} \mid o_{1:t}, z_t)$ which predicts the next observation given the latent

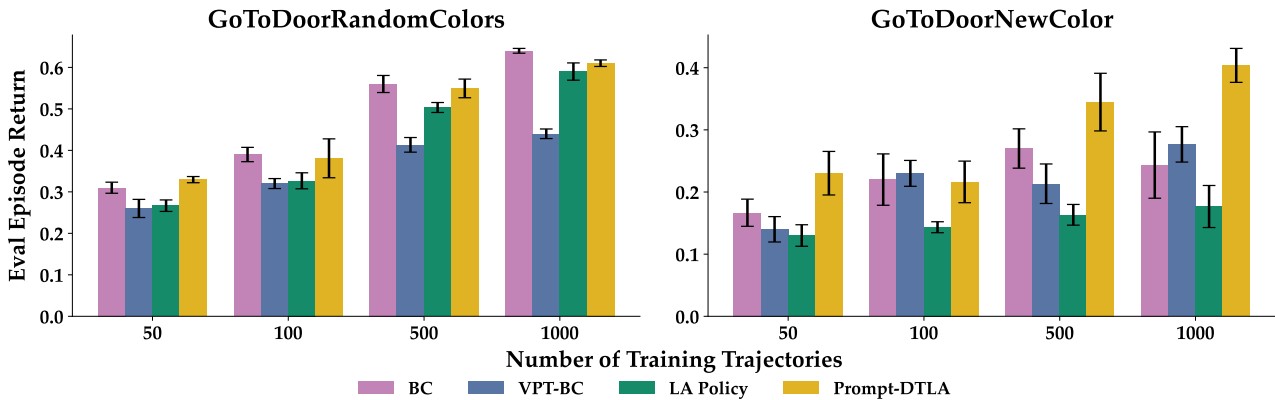

Figure 2: Evaluation results in two different task settings. **(Left)** Agent is trained in `GoToDoorRandomColors` and evaluated in the same domain. **BC** performs well as it is trained on action-labelled trajectories, while the other methods are trained on *unlabelled* data. **LA Policy** outperforms **VPT-BC** which is trained on actions labeled using an IDM. Both **LA Policy** and **Prompt-DTLA** scale with more data, performing similarly to **BC**. **(Right)** Agent is trained in `GoToDoorRandomColors` and evaluated in a new domain, `GoToDoorNewColor`. **Prompt-DTLA** can utilize the in-context demonstration to generalize to *unseen* tasks, outperforming all of the baselines by a significant margin as the number of unlabelled expert trajectories increases. Error bars represent standard error computed across three random seeds.

action and the previous observations. We use the IDM to label sequences of videos for training and rollout of the transformer. The FDM provides the learning signal to train the IDM and is discarded at test time. At test time, our environment expects a particular action space, which will likely not be identical to the learned latent action space. Thus, to map latent actions to environment actions, we train an action decoder, $p_\omega(a \mid z)$ using a small action-labeled dataset of transitions.

**Prompt-DTLA Training.** We are interested in few-shot generalization to new tasks after pretraining on a large offline unlabelled dataset. Our model should learn to condition on a small set of few-shot examples to understand the context of which MDP it is in and autoregressively predict the actions conditioned on the provided context. Following prior literature in natural language processing (Min et al., 2022) and sequential decision-making (Raparthy et al., 2023), we concatenate multiple training trajectories into one long sequence of tokens and input this into a Causal Transformer model (Chen et al., 2021) (see Figure 1). We assume our trajectories come from expert demonstrators. Each trajectory sequence is defined as $\tau_i = (o_1, z_1, o_2, z_2, \ldots, o_T, z_T)$, where consecutive observations $(o_t, o_{t+1})$ have been annotated with latent actions $z_t$ by our pretrained LAM. We concatenate together $K$ such trajectory sequences of the same MDP. The difference between the trajectories is in the initial state of the agent. Thus, $\tau^{input} = (\tau_1, \tau_2, \ldots, \tau_K)$. Since each timestep is a 2-tuple of $(o, z)$, the input sequence to the Transformer corresponds to $2KT$ tokens. Following (Chen et al., 2021), we use linear layers to embed the observation tokens and use the timestep as the

positional encoding. Prompt-DTLA autoregressively predicts $KT$ latent actions at heads corresponding to the state tokens in the input sequence. Our training objective is, $\mathcal{L}_{MSE} = \sum_{t=1}^{T} ||z_t - \hat{z}_t||_2^2$. We summarize the full algorithm for training Prompt-DTLA in Algorithm 1.

**Prompt-DTLA Inference:** We evaluate the trained policy on *unseen* tasks different from the ones used in training. We perform one-shot adaptation by concatenating an expert trajectory prompt, $\tau^{prompt} = (o_1, f_\phi(o_1, o_2), o_2, \ldots, f_\phi(o_{T-1}, o_T), o_T)$, with the execution history $\tau_{:t} = (o_1, z_1, \ldots, o_{t-1}, z_t)$. We use the pretrained LAM, $f_\phi$, to label the latent actions between the observations of the expert demonstration. The input to the Transformer is, $\tau^{input} = (\tau^{prompt}, \tau_{:t})$ and it predicts the next action, $p(a_{t+1} \mid o_{t+1}, \tau^{input})$. The full inference loop is summarized in Algorithm 2.

## 4. Experiments

We investigate the in-context RL capabilities of Prompt-DTLA. We present experiments in XLand-Minigrid (Nikulin et al., 2023), a JAX-based implementation of Minigrid (Chevalier-Boisvert et al., 2023) discrete maze environments. The agent is a red triangle randomly initialized in an $N \times N$ 2D grid. Each task is defined by a goal and a set of rules which describe the environment dynamics and reward function. We design a custom task `GoToDoorRandom`, shown in Figure 3. Four doors are located at the middle of each wall and a ball at the center of the room determines which color door the agent should navigate to. There are six discrete actions in XLand-Minigrid: forward, turn clock-

| Num labelled trajs | VPT | Action Decoder |
|---|---|---|
| 10 | $0.81 \pm 0.01$ | $0.97 \pm 0.01$ |
| 15 | $0.82 \pm 0.01$ | $0.97 \pm 0.01$ |
| 20 | $0.84 \pm 0.01$ | $0.98 \pm 0.01$ |
| 50 | $0.91 \pm 0.02$ | $0.99 \pm 0.01$ |
| 75 | $0.95 \pm 0.02$ | $1.00 \pm 0.00$ |

Table 1: Action decoding accuracy between IDM and LAM using the same amount of labelled trajectories. We report mean and standard error over three seeds.

wise, turn counterclockwise, pick up, put down, and unlock door. We train an oracle policy using goal-conditioned PPO (Schulman et al., 2017) and collect an offline dataset of 10k expert trajectories. We use the entire dataset for training the latent action model and randomly sample trajectories to train the action decoder.

### 4.1. Baselines

The objective of this work is to investigate different methods for learning a policy from unlabelled video data and provide insights into the importance of different design choices.

**Behavior Cloning (BC):** Standard BC with an MLP policy using $N$ action-labelled trajectories.

**Video-Pretraining (VPT-BC) (Baker et al., 2022) Policy:** Train an Inverse Dynamics Model following (Baker et al., 2022) using $N$ action-labelled trajectories. We use the pretrained IDM to annotate the remainder of the transitions in the offline dataset for training an MLP policy via BC.

**Latent Action Policy (LA Policy) (Schmidt & Jiang, 2023):** Standard BC with an MLP policy on observation-only dataset with latent actions annotated using a pretrained LAM. This baseline does not perform in-context learning and will suffer when there is task ambiguity.

**Insight #1: Learning a LAM is more data-efficient than IDM resulting in better evaluation performance.** We explore two approaches for annotating offline data with actions, either learning an IDM or a LAM. The supervision signal for IDM training is limited to only a small action-labelled dataset. In contrast, the LAM training objective is unsupervised and can utilize the full offline dataset. In Table 1, we compare the ground-truth action decoding accuracy between IDM and LAM with the same amount of labelled data, demonstrating that learning a LAM is indeed more data-efficient. Our action decoder achieves 97% prediction accuracy with only 10 labelled demonstrations, while an IDM trained with the same amount of data achieves only 81%. Using a more accurate model for annotating our offline data results in a better performing downstream policy. In Tables 3 and 4, we observe that VPT-BC performance

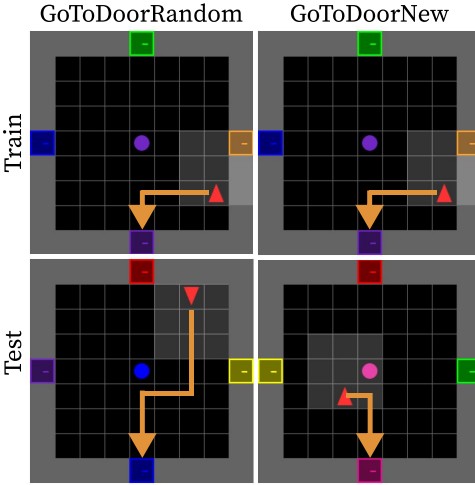

Figure 3: **XLand-Minigrid** tasks: `GoToDoorRandom` and `GoToDoorNew`. The initial location of the agent is randomized at the start of each episode. The door locations are fixed during training and a new color (pink) door is introduced during evaluation time to measure generalization.

suffers from an inaccurate IDM and does not scale as well as LA Policy with more data.

**Insight #2: Prompt-DTLA generalizes to unseen tasks with few-shot expert demonstrations using in-context learning.** To measure the generalization capabilities of Prompt-DTLA, we evaluate each method on a new, *unseen* task, `GoToDoorNew` shown in Figure 3, which introduces a pink door not present during training. In Figure 2, we observe that Prompt-DTLA outperforms each of the baselines on this generalization task, with over 40% improvement over the next best baseline and scales with more unlabelled expert trajectories. Prompt-DTLA leverages the one-shot expert demonstration of the task to learn the correct behavior and ascertain the task information.

## 5. Future Work

For future work, we plan to evaluate Prompt-DTLA in more challenging, realistic domains including Procgen (Cobbe et al., 2020) and Overcooked AI (Carroll et al., 2019). We are also interested in extending Prompt-DTLA to learn latent action models for continuous action spaces. One exciting application of our approach is few-shot generalization of robotic control tasks from pretraining on internet-scale videos. We aim to conduct more comprehensive analysis to investigate: 1) how to most effectively learn a useful latent action space, 2) how Prompt-DTLA scales with more offline data across many tasks, and 3) whether Prompt-DTLA can generalize under different degrees of distribution shift between training and test?

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

# A. Appendix / supplemental material

## A.1. Hyperparameters

We provide training and modeling hyperparameters for Prompt-DTLA in Table 2.

Table 2: Hyperparameters for models in Prompt-DTLA

|  | Hyperparameter | Value |
|---|---|---|
| Image Encoder / Decoder | channels | [16, 32, 32, 64] |
|  | use batch norm | True |
|  | kernel sizes | [3, 3, 2, 2] |
|  | stride | 1 |
|  | padding | VALID |
| Latent Action Model | embedding dim | 256 |
|  | $\beta$ | 0.05 |
|  | commitment loss weight | 1.0 |
|  | ema decay | 0.999 |
|  | num latent codes | 60 |
|  | code dim | 64 |
|  | context len | 1 |
|  | mlp sizes | [128, 128] |
|  | num gradient updates | 50000 |
|  | lr | 3e-4 |
|  | lr | linear-constant schedule |
|  | lr warmup steps | 10000 |
| Action Decoder | mlp sizes | [128, 128] |
|  | lr | 3e-4 |
|  | num gradient updates | 10000 |
| Transformer | number of stitched trajectories $K$ | 2 |
|  | batch size | 128 |
|  | num eval rollouts | 50 |
|  | learning rate | 3e-4 |
|  | layer sizes | [128, 128] |
|  | embedding dimension | 128 |
|  | num layers | 1 |
|  | num attention head | 3 |
|  | activation | gelu |
|  | dropout | 0.1 |
|  | num gradient updates | 50000 |

For image observations, we use a U-net architecture following (Schmidt & Jiang, 2023). The encoder consists of 4 downsampling blocks. Each downsampling block is a convolutional layer followed by a batch norm, a residual block (Espeholt et al., 2018) and a GELU activation. The residual block applies two convolution layers with kernel size 3 and same padding. The decoder consists of 4 upsampling blocks which comprises of a ConvTranspose layer followed by batch norm, residual block, and GELU activation.

## A.2. Additional Results / Analysis

We provide additional experimental results in Table 5. We evaluate the policy 20 times over 25,000 gradient update steps. For each evaluation step, we perform 50 rollouts and average the final episode return. We report the highest average episode return across all evaluation checkpoints for each method with standard deviation over three random seeds. We evaluate each method on a holdout set of tasks separate from the training data.

| Num labelled trajectories | Num unlabelled trajectories | GoToDoorRandom |
|---|---|---|
| 10 | 50 | $0.24 \pm 0.02$ |
| | 100 | $0.32 \pm 0.03$ |
| | 500 | $0.46 \pm 0.01$ |
| | 1000 | $0.53 \pm 0.02$ |
| 20 | 50 | $0.26 \pm 0.02$ |
| | 100 | $0.32 \pm 0.01$ |
| | 500 | $0.41 \pm 0.02$ |
| | 1000 | $0.44 \pm 0.01$ |

Table 3: VPT-BC evaluation results. Ablation study over different amounts of labelled data for learning the IDM and number of unlabelled trajectories for training the policy. We report mean and standard error over three random seeds.

| Num labelled trajectories | Num unlabelled trajectories | GoToDoorRandom |
|---|---|---|
| 10 | 50 | $0.25 \pm 0.04$ |
| | 100 | $0.34 \pm 0.02$ |
| | 500 | $0.56 \pm 0.01$ |
| | 1000 | $0.63 \pm 0.02$ |
| 20 | 50 | $0.26 \pm 0.03$ |
| | 100 | $0.36 \pm 0.04$ |
| | 500 | $0.50 \pm 0.01$ |
| | 1000 | $0.59 \pm 0.02$ |

Table 4: Latent Action Policy evaluation results. Ablation study over different amounts of labelled data for learning the IDM and number of unlabelled trajectories for training the policy. We report mean and standard error over three random seeds.

| Num of train (unlabelled) trajs | Method | GoToDoorRandom | GoToDoorNew |
|---|---|---|---|
| 50 | BC | $0.31 \pm 0.01$ | $0.17 \pm 0.02$ |
| | VPT-BC | $0.26 \pm 0.02$ | $0.14 \pm 0.02$ |
| | Latent Action Policy | $0.26 \pm 0.03$ | $0.09 \pm 0.03$ |
| | Prompt-DTLA | $\mathbf{0.33 \pm 0.01}$ | $\mathbf{0.23 \pm 0.04}$ |
| 100 | BC | $\mathbf{0.39 \pm 0.02}$ | $0.22 \pm 0.04$ |
| | VPT-BC | $0.32 \pm 0.01$ | $0.23 \pm 0.02$ |
| | Latent Action Policy | $0.33 \pm 0.02$ | $0.13 \pm 0.00$ |
| | Prompt-DTLA | $0.38 \pm 0.05$ | $\mathbf{0.22 \pm 0.03}$ |
| 500 | BC | $\mathbf{0.56 \pm 0.02}$ | $0.27 \pm 0.03$ |
| | VPT-BC | $0.41 \pm 0.02$ | $0.21 \pm 0.03$ |
| | Latent Action Policy | $0.52 \pm 0.03$ | $0.17 \pm 0.02$ |
| | Prompt-DTLA | $0.55 \pm 0.02$ | $\mathbf{0.34 \pm 0.05}$ |
| 1000 | BC | $\mathbf{0.64 \pm 0.01}$ | $0.24 \pm 0.05$ |
| | VPT-BC | $0.44 \pm 0.01$ | $0.28 \pm 0.03$ |
| | Latent Action Policy | $0.59 \pm 0.01$ | $0.22 \pm 0.04$ |
| | Prompt-DTLA | $0.61 \pm 0.01$ | $\mathbf{0.40 \pm 0.03}$ |

Table 5: Evaluation results of Prompt-DTLA and baseline methods on two environments. Action Decoder and IDM are trained with 20 labelled trajectories. We report mean and standard error over three random seeds.

## A.3. Environments

We conduct experiments in the XLand-Minigrid (Nikulin et al., 2023) environment, a JAX-based implementation of Minigrid (Chevalier-Boisvert et al., 2023) 2D maze environments. The agent is a red triangle randomly initialized an $9 \times 9$ 2D grid. Each task is defined by a goal and a set of rules which describe the environment dynamics and reward function. We design a new task `GoToDoorRandom` and a variant of this task `GoToDoorNew`, shown in Figure 3. In `GoToDoorRandom`, a door is located at the middle of each of the four walls and a ball in the center of the room determines which color door the agent should navigate to. The door colors are randomly sampled from a set of six colors: $\{red, green, blue, yellow, orange, purple\}$. In `GoToDoorNew`, a random door is made $pink$. The agent's observation is a $9 \times 9 \times 3$ symbolic array of the entire grid. We do not use partial observation. The first channel represents tile, the second represents color, and the last channel contains the agent's heading. There are six discrete actions: forward, turn clockwise, turn counterclockwise, pick up, put down, and unlock door. The episode terminates either when the target goal is reached or when the max episode length of 30 is reached.

## A.4. Algorithms

---

**Algorithm 1** Prompt-Decision Transformers with Latent Actions (Prompt-DTLA)

---

1: **Input:** $\mathcal{D}_{unlabelled}$, $\mathcal{D}_{labelled}$, IDM $f_\phi$, FDM $g_\psi$, Action decoder $p_\omega$, $Transformer_\theta$, training tasks $\mathcal{T}_{train}$
2: **for** iter = 1 to num_lam_training_steps **do**
3:     Sample training example $(o_1, \ldots, o_t, o_{t+1})$ from $\mathcal{D}_{unlabelled}$                {train LAM}
4:     $h_t = f_\phi(\cdot \mid o_1, \ldots, o_t, o_{t+1})$
5:     $z_t = VectorQuantize(h_t)$
6:     $\hat{o}_{t+1} = g_\psi(\cdot \mid o_1, \ldots, o_t, z_t)$
7:     $\mathcal{L}_{MSE} = ||o_{t+1} - \hat{o}_{t+1}||_2^2$, Update $\phi$ and $\psi$
8: **end for**
9: **for** iter = 1 to num_decoder_training_steps **do**
10:     Sample training example $(o_1, a_1, o_2, \ldots, a_t, o_{t+1})$ from $\mathcal{D}_{labelled}$                {train action decoder}
11:     Predict latent action with LAM $z_t = f_\phi(\cdot \mid o_1, \ldots, o_t, o_{t+1})$
12:     $\hat{a}_t = p_\omega(\cdot \mid z_t)$
13:     $\mathcal{L}_{dec} = CrossEntropy(a_t, \hat{a}_t)$, Update $\omega$
14: **end for**
15: **for** iter = 1 to num_policy_training_steps **do**
16:     Sample training task $\mathcal{T}_i \sim \mathcal{T}^{train}$                {train ICL policy}
17:     Sample $k$ demos from $\mathcal{D}_{unlabelled}$ for task $\mathcal{T}_i$
18:     Label demos with latent actions: $\tau_i^* = (o_1, f_\phi(o_1, o_2), o_2, \ldots, f_\phi(o_{T-1}, o_T), o_T)$
19:     Construct multi-trajectory input $\tau = (\tau_1, \ldots, \tau_k)$
20:     Predict latent action for each $t$ in each $\tau_i$, $\hat{z}_{1:kT} = Transformer_\theta(\tau)$
21:     $\mathcal{L}_{MSE} = ||z_{1:kT} - \hat{z}_{1:kT}||_2^2$, Update $\theta$
22: **end for**

---

**Algorithm 2** Prompt-DTLA Few-Shot Inference

---

1: **Input:** test tasks $\mathcal{T}^{test}$, expert demos $\tau^*$, IDM $f_\phi$, Action decoder $p_\omega$
2: Sample trajectory prompt(s) from $\tau^*$ for task $\mathcal{T}_i \sim \mathcal{T}^{test}$, $\tau_i^* = (o_1, o_2, \ldots, o_T)$
3: Label prompt with latent actions: $\tau_i^* = (o_1, f_\phi(o_1, o_2), o_2, \ldots, f_\phi(o_{T-1}, o_T), o_T)$
4: Initialize current trajectory history, $\tau = \{\}$
5: **while** not done **do**
6:     $z = Transformer_\theta((\tau^*, \tau))[-1]$
7:     $a = p_\omega(\cdot \mid z)$
8:     $o'$, done = env_step($a$)
9:     Append $o'$, $z$ to trajectory history $\tau$
10: **end while**

---

**A.5. Computational Resources**

All experiments can be run on a single Nvidia RTX A6000 GPU. The following are rough estimates of average run-time for the XLand-Minigrid experiments.

- BC: 1 hour
- Latent Action Model Training: 30 minutes
- Action Decoder Training: 20 minutes
- Prompt-DTLA: 1 hour

