# OpenReview forum: "In-Context Generalization to New Tasks From Unlabeled Observation Data"
_ICML.cc/2024/Workshop/ICL — ICML 2024 Workshop ICL Poster_

### Official Review · Reviewer_UpJX · 2024-06-04
**Interesting paper of in-context meta-reinforcement learning**

**Rating:** 2
**Fit:** 2
**Confidence:** 3

**Workshop Review:**

This paper is clearly written and easy to follow. The authors propose a label-free meta-reinforcement learning framework that enables adaptation to new tasks through in-context learning. Specifically, they developed a latent action model to recover latent actions from observations. During the inference stage, this model is used to label expert demonstrations. The novelty of this work lies in its label-free pre-training strategy. This study is well-aligned with the interests of the ICL workshop.

**Reason For Not Giving Higher Score:**

1. The proposed method is only evaluated on simple grid environments. It is interesting and important to evaluate its effectiveness on more complicated environments such as MuJoCo and Meta-World.
2. The improvement of the proposed method is marginal on the GoToDoorRandomColors (slightly worse than behavior cloning).

**Reason For Not Giving Lower Score:**

1. The paper is clearly written. The proposed method is technically sound to me.
2. The experimental results are somewhat promising, but more extensive evaluation is needed to understand the effectiveness of the proposed method.

---

### Official Review · Reviewer_zqvH · 2024-06-08
**A paper attempting to do ICL for policy generalization, but demonstrates only a small generalization in a toy domain.**

**Rating:** 2
**Fit:** 3
**Confidence:** 2

**Workshop Review:**

This paper is an attempt to use unlabeled videos and in-context learning for policy generalization. The idea is to first train a latent action model for unlabeled video sequences. An action predictor is trained to map the latent action to real action. A causal transformer is auto regressively trained to predict just the latent actions in a sequence of alternating observations and latent-actions, where the latent-actions in the input are filled by the previously trained latent-action-model. The claim is that this mechanism generalizes better due to ICL in the transformer.

While this is an interesting idea, the experiments are too small scale to demonstrate this convincingly. For example, what would happen if the door location was changed, not just the door color? Is this part of the tests? I couldn't figure out from a quick read. What if all the colors are new? Also, 40% success on this task looks rather small. Ablations would help to understand why this is so.

Nevertheless I think this is an interesting contribution for a workshop.

**Reason For Not Giving Higher Score:**

Reasons are given in the review above.

While this is an interesting idea, the experiments are too small scale to demonstrate this convincingly. For example, what would happen if the door location was changed, not just the door color? Is this part of the tests? I couldn't figure out from a quick read. What if all the colors are new? Also, 40% success on this task looks rather small. Ablations would help to understand why this is so.

**Reason For Not Giving Lower Score:**

The paper is reasonably well written and explains the idea properly.

---

### Meta-Review · Area_Chair_eV1F · 2024-06-16

**Recommendation:** 2

**Metareview:**

The paper presents a novel label-free meta-reinforcement learning framework that leverages in-context learning to adapt to new tasks, utilizing a latent action model to recover actions from observations. There are some concerns about the scale and comprehensiveness of the experiments but the reviewers acknowledge the innovative approach and its alignment with the workshop's focus. The clear writing and the potential of the label-free pre-training strategy to enhance policy generalization make this paper a valuable contribution to the ICL workshop. Therefore, it is recommended for acceptance.

---

### Decision · Program_Chairs · 2024-06-17

**Decision:**

Accept (Poster)

**Comment:**

**Accept with minor revision**: Please clarify what "video" refers to, as the setting appears to be navigation in a GridWorld environment.